# Universally high transcript error rates in bacteria

**Weiyi Li[1†], Michael Lynch[1,2]\***

[1]Department of Biology, Indiana University, Bloomington, United States; [2]Center for Mechanisms of Evolution, The Biodesign Institute, Arizona State University, Tempe, United States

**Abstract** Errors can occur at any level during the replication and transcription of genetic information. Genetic mutations derived mainly from replication errors have been extensively studied. However, fundamental details of transcript errors, such as their rate, molecular spectrum, and functional effects, remain largely unknown. To globally identify transcript errors, we applied an adapted rolling-circle sequencing approach to *Escherichia coli*, *Bacillus subtilis*, *Agrobacterium tumefaciens*, and *Mesoplasma florum,* revealing transcript-error rates 3 to 4 orders of magnitude higher than the corresponding genetic mutation rates. The majority of detected errors would result in amino-acid changes, if translated. With errors identified from 9929 loci, the molecular spectrum and distribution of errors were uncovered in great detail. A G→A substitution bias was observed in *M. florum*, which apparently has an error-prone RNA polymerase. Surprisingly, an increased frequency of nonsense errors towards the 3′ end of mRNAs was observed, suggesting a Nonsense-Mediated Decay-like quality-control mechanism in prokaryotes.

**\*For correspondence:**
mlynch11@asu.edu

**Present address:** [†]Joint Initiative for Metrology in Biology, SLAC National Accelerator Laboratory, Stanford University, Stanford, United States

**Competing interests:** The authors declare that no competing interests exist.

## Introduction

Transcript errors refer to any inconsistencies between RNA transcripts and their corresponding genomic loci. They can occur during ribonucleotide (rNTP) incorporations by RNA polymerases and/or via post-transcriptional modifications. Errors on RNA transcripts may directly cause dysfunctions due to the regulatory roles of small RNAs and the fate determination of mRNAs by RNA structural motifs (*Strathern et al., 2012*). Such errors can also indirectly induce various effects at the protein level. Transcript errors can inactivate proteins and result in a loss-of-function (*Gordon et al., 2013*). They can also indirectly give rise to misfolded proteins and induce proteotoxic stress (*Gout et al., 2017*; *Vermulst et al., 2015*). Errors on RNA transcripts may be causal factors leading to neuron degenerative diseases (*van Leeuwen et al., 1998a*; *van Leeuwen et al., 1998b*) and tumorigenesis (*Saxowsky et al., 2008*). Therefore, transcript errors represent a significant potential mechanism influencing cellular integrity and fitness.

Reporter-construct assays have long been the major approach to evaluating the fidelity of RNA polymerases and identifying transcript errors (*Blank et al., 1986*; *Bubunenko et al., 2017*; *Nesser et al., 2006*; *Rosenberger and Foskett, 1981*; *Rosenberger and Hilton, 1983*; *Shaw et al., 2002*; *Springgate and Loeb, 1975*; *Strathern et al., 2012*), but these methods focus only on individual loci and cannot identify errors without phenotypic marker effects. Conventional high-throughput sequencing approaches have been considered to identify transcript errors at a large scale (*van Dijk et al., 2015*). However, the challenge is to distinguish the real signal of transcript errors from noise produced by technical errors resulting during reverse transcription and sequencing. To circumvent this problem, a rolling-circle amplification-based sequencing (CirSeq) method (*Acevedo and Andino, 2014*; *Acevedo et al., 2014*; *Lou et al., 2013*) was recently proposed and later applied to identify transcript errors in the whole transcriptome of prokaryotes (*Traverse and*

**eLife digest** Most cells contain molecules of DNA that carry instructions to make the proteins cells need to perform different tasks. When a cell requires a certain protein, the corresponding DNA sequence is first transcribed into molecules of ribonucleic acid (RNA) known as transcripts. These sequences of RNA are then read by the cell and translated into the desired protein sequence.

Errors in copying DNA before a cell divides, can lead to genetic mutations that affect the ability of the cell to carry out certain roles, influencing the overall 'fitness' of the cell. Similar to genetic mutations, errors that arise when forming RNA transcripts may also alter the tasks a cell performs. However, it is difficult to find out what kinds of errors cells have in their transcripts and how often these mistakes occur. This is because current methods for sequencing RNA are prone to technical inaccuracies that interfere with the ability to detect true transcript errors.

Now, Li and Lynch have adapted a method for high-throughput sequencing of RNA, which can accurately identify transcript errors in *Escherichia coli* and other species of bacteria. The experiments showed that errors in RNA molecules occurred more frequently than genetic mutations in the same sequence of DNA. Li and Lynch also found that the transcripts contained more nonsense errors – that is, mutations which prematurely stop transcripts from being translated, resulting in shorter proteins – at the end of the RNA molecule than at the beginning or middle. It is possible that transcripts with errors at the beginning or the middle are more efficiently eliminated than those at the end, suggesting that bacteria have a quality-control mechanism for removing transcripts with premature stop sequences.

These findings suggest that at any one-time cells carry thousands of transcripts with inaccuracies in their sequence, which likely impact the tasks cells perform. The next step will be to investigate how these different transcript errors affect the fitness of cells.

*Ochman, 2016*). We further modified this protocol to minimize RNA damage potentially introduced during the preparation of sequencing libraries (*Gout et al., 2017*).

In this study, we applied an adapted CirSeq approach, which has been demonstrated to identify transcript errors accurately and efficiently at a large scale in eukaryotes (*Gout et al., 2017*), to prokaryotes for the first time. A large number of transcript errors was detected, and transcript-error rates were revealed to be orders of magnitude higher than corresponding genetic mutation rates. Our results indicate that the bias in molecular spectra of transcript errors can be influenced by both RNA polymerases and cellular rNTP concentrations. Furthermore, the spatial distribution of transcript errors on RNAs provides novel insights into the mechanism of RNA quality-control in prokaryotes.

## Results

### A global view of the transcript error distribution

Applying the adapted CirSeq method (see Materials and methods) to *E. coli*, *B. subtilis*, *A. tumefaciens*, and *M. florum*, RNA sequencing libraries were made with three biological replicates for each species. Key steps of library preparations involve circularizing RNA fragments and generating cDNAs with tandem repeats by rolling-circle reverse transcription. In this way, transcript errors tend to appear on all repeats of sequencing reads, while sequencing and reverse transcription errors are nearly always revealed as singletons (*Figure 1—figure supplement 1*). The number of loci where transcript errors were identified from each species ranges from 2006 to 2942, totaling 9929 loci across all species. *M. florum* showed a per-site error rate of $1.82 \pm 0.01$ (SEM) $\times 10^{-5}$, the highest among the four species ($P = 0.009$, Mann-Whitney U test). The error rates in *E. coli*, *B. subtilis*, and *A. tumefaciens* were $5.84 \pm 0.10$ (SEM) $\times 10^{-6}$, $5.80 \pm 0.14$ (SEM) $\times 10^{-6}$, and $7.26 \pm 0.35$ (SEM) $\times 10^{-6}$, respectively. These error rates are 3 to 4 orders of magnitude higher than the corresponding genomic (DNA-level) mutation rates estimated from mutation-accumulation experiments in these species (*Lee et al., 2012*; *Lynch et al., 2016*; *Sung et al., 2016*; *Sung et al., 2015*; *Sung et al., 2012*).

With such a large number of transcript errors identified, a transcriptome-wide view of the error distribution in each species was uncovered. Based on the circular genomes of bacteria (except for *A. tumefaciens*, which has one circular chromosome, one linear chromosome, and two plasmids [*Goodner et al., 2001*]), we annotated genomic positions of transcript errors with different potential functional effects and plotted transcript-error rates in 10 kb sliding windows (1 kb for *M. florum*) (*Figure 1*). To test whether transcript errors are randomly distributed across different genes, a previously proposed test (*Long et al., 2016*) was performed to identify genes enriched with transcript errors. For each gene, the expected number of transcript errors was calculated as the product of the average transcriptome-wide error rate per base and the sequencing coverage of the gene. The Poisson probability of observing a number of errors greater than or equal to the observed number was calculated. Out of 607, 495, 586, and 186 genes with detected transcript errors in *E. coli*, *A. tumefaciens*, *B. subtilis* and *M. florum*, respectively, 1, 4, 0 and 4 genes were revealed to have significantly larger numbers of errors than random expectations (Bonferroni-corrected *P* values of 0.05, *Supplementary file 1*, Tables 2-5), suggesting that transcript errors are in general randomly distributed across genes.

The whole bacterial transcriptome is synthesized by a single type of RNA polymerase. However, RNA products from protein-coding and noncoding RNA (ncRNA) regions undergo distinct co- and post-transcriptional processes. mRNAs are mature upon transcription and ready for translation, while ncRNAs, such as ribosomal RNAs (rRNA) and transfer RNAs (tRNA), need to be further processed to be functional (*Cooper, 2000*). To evaluate whether transcript-error rates of these two genomic regions are different, we calculated the error rates of protein-coding and ncRNA transcripts by dividing the number of errors by the number of nucleotides assayed in corresponding regions. Transcript-error rates of these two regions are similar in *E. coli* and *A. tumefaciens*, but the error rate of ncRNA transcripts is higher than that of protein-coding transcripts in *B. subtilis* and lower in *M. florum* (p<0.05, paired t-test) (*Figure 2*).

## The molecular spectra of transcript errors are biased to C→U and G→A substitutions

A transition/transversion bias of genetic mutations has been widely observed in different species, with the molecular spectrum mostly dominated by G:C→A:T substitutions (*Hershberg and Petrov, 2010*; *Hildebrand et al., 2010*; *Lynch, 2010*). However, knowledge on the molecular spectrum of transcript errors in prokaryotes remains limited (*Imashimizu et al., 2015*; *Traverse and Ochman, 2016*; *Traverse and Ochman, 2018*). In this study, we calculated the error rate of all twelve categories of substitutions for each species (*Figure 3*), revealing a general bias of transitions over transversions. This bias has been thought to be driven solely by C→U substitutions (*Traverse and Ochman, 2016*), which may mainly result from post-transcriptional cytosine deaminations. However, the transition/transversion bias here even holds after C→U substitutions are excluded ($P < 0.005$, $\chi^2$ test, *Supplementary file 1*, Table 6). This observation indicates that the transcriptional machinery in bacteria, similar to the replication machinery, tends to have a low ability to distinguish rNTPs within the same structural class of nitrogenous bases (*Keightley et al., 2009*; *Kucukyildirim et al., 2016*; *Lee et al., 2012*; *Long et al., 2015a*; *Long et al., 2015b*; *Lynch, 2007*; *Lynch, 2010*; *Lynch et al., 2008*; *Ossowski et al., 2010*; *Sung et al., 2015*). Of all transitions, the C→U substitution rate is consistently high in all four species. In addition, an unexpectedly high G→A substitution rate is revealed in *M. florum*, which displayed the highest transcript-error rates among four species in the present study. Intriguingly, this substitution bias was also recently observed in yeast and *E. coli* transcription-machinery mutants with decreased fidelity (*Gout et al., 2017*; *Imashimizu et al., 2015*; *Traverse and Ochman, 2018*). Thus, the G→A substitution bias may be a signature of error-prone RNA polymerase in both eukaryotes and prokaryotes.

## Characterization of transcript errors

To evaluate potential functional effects of transcript errors, we categorized transcript errors within protein-coding regions into synonymous, missense, and nonsense substitutions using SnpEff (*Cingolani et al., 2012*; *Table 1*). Based on the bias of rNTP substitution rates (*Figure 3*) and codon usages of each bacterium, we also calculated the expected percentages of each error type under the assumption that transcript errors are randomly generated across the genome without error-

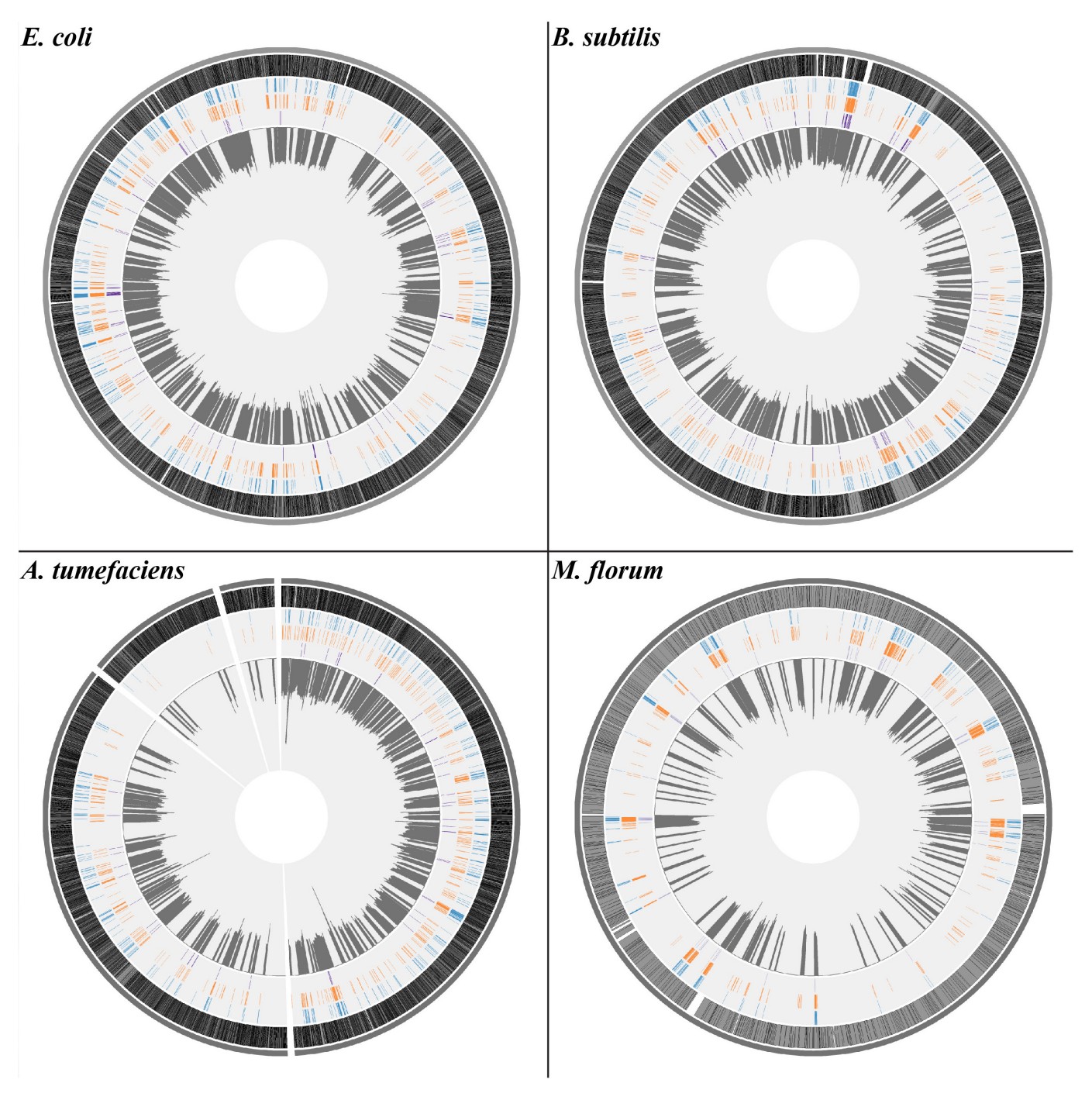

**Figure 1.** The distribution of transcript errors across the whole transcriptomes of *E. coli*, *B. subtilis*, *A. tumefaciens*, and *M. florum*. The first nucleotide of the circular chromosome starts at the 12 o'clock position. For *A. tumefaciens*, chromosomes and plasmids are arranged from the largest to smallest size in a clockwise orientation. From the outer ring to the inner ring: bacterial chromosomes (dark gray), protein-coding region (grey, black strokes indicate gene densities), synonymous substitutions (blue), missense substitutions (orange), nonsense substitutions (purple) and average transcript-error rates (plots in dark gray) in a 10 kb sliding window with a step size of 1 bp (1 kb windows for *M. florum*). Windows without sufficient sequencing coverages to detect transcript errors are left blank.

The online version of this article includes the following source data and figure supplement(s) for figure 1:

**Source data 1.** Numerical data that are represented as a graph in *Figure 1*.

**Figure supplement 1.** The flowchart of CirSeq method.

**Figure supplement 2.** An overview of the bioinformatic pipeline to process CirSeq reads to identify transcript errors.

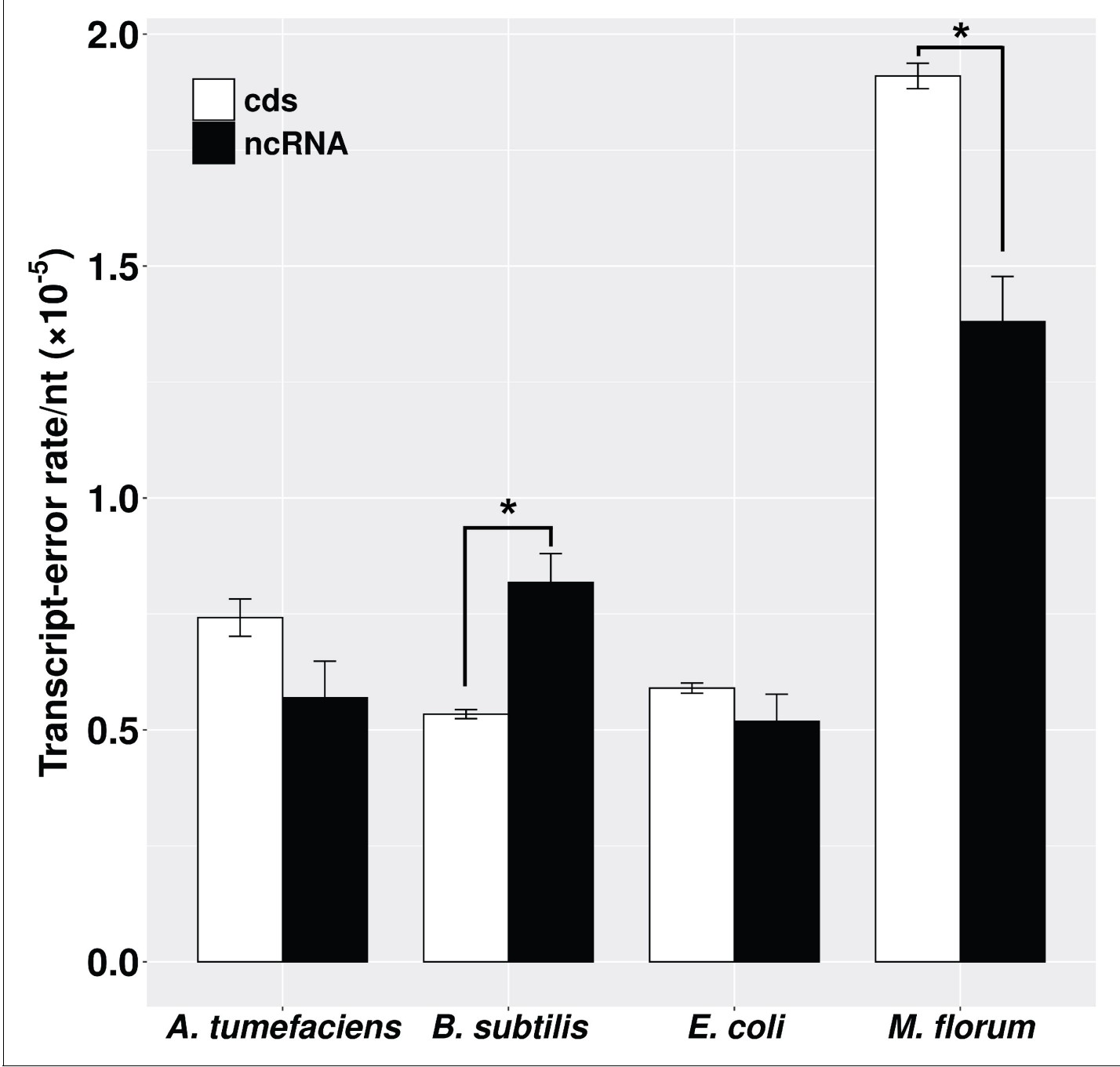

**Figure 2.** Transcript-error rates of protein-coding and ncRNA regions. cds includes all protein-coding genes that were sequenced in this study. ncRNA refers to RNAs that are functional but not translated into proteins, for example tRNA and rRNA. Transcript-error rates were calculated by dividing the number of errors by the number of nucleotides assayed in corresponding regions. Error bars indicate standard errors. The level of significance difference is indicated by asterisks (*p<0.05, paired t-test).

The online version of this article includes the following source data for figure 2:

**Source data 1.** Numerical data that are represented as a graph in *Figure 2*.

correction processes (see Materials and methods, and *Supplementary file 1*, Table 7). Consistent with observations, the majority of transcript errors are expected to result in amino-acid changes, if translated (*Table 1*). For nonsense errors, the observed percentages are close to or significantly lower than the random expectation ($P < 0.005$, $\chi^2$ test, *Table 1*).

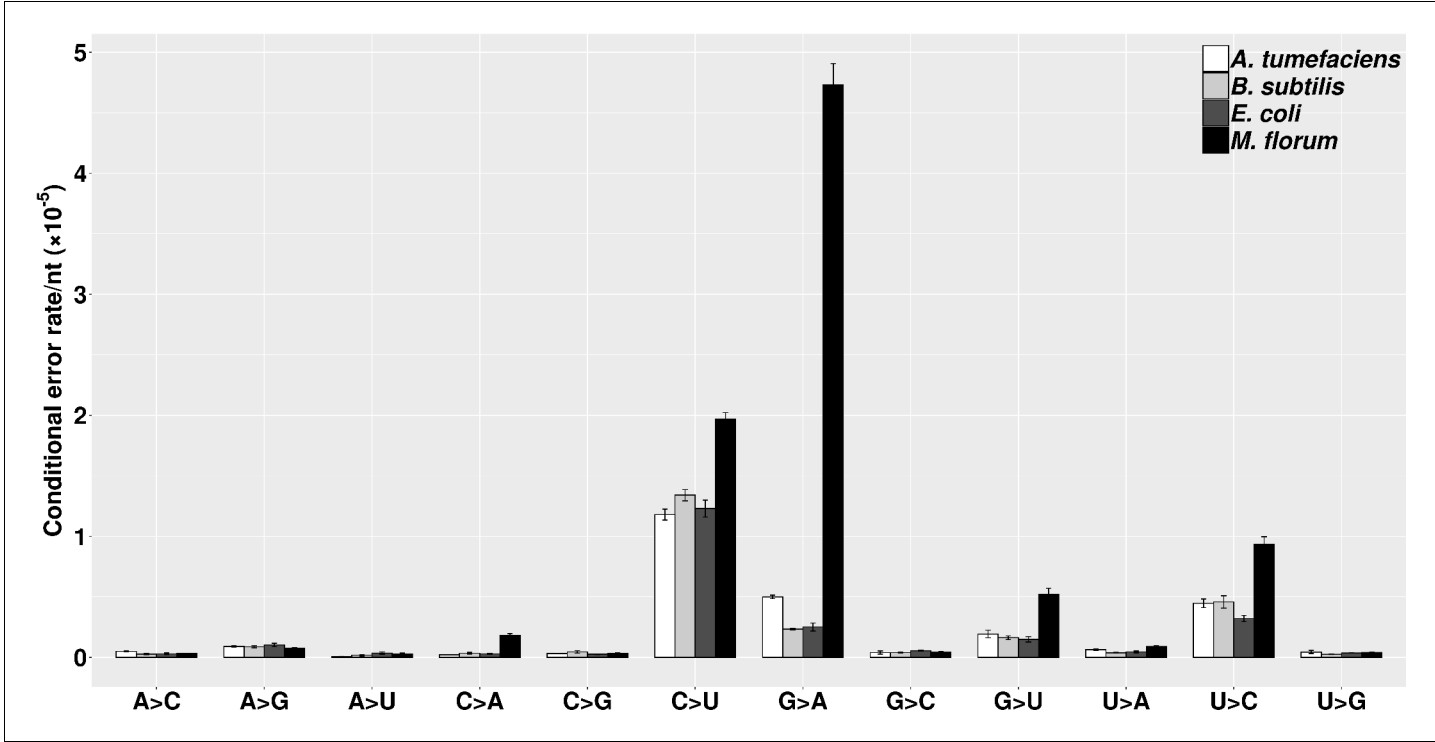

**Figure 3.** The molecular spectra of transcript errors for four bacterial species. The conditional error rates of each type of substitutions were calculated from the number of particular transcript errors, divided by the number of corresponding ribonucleotides assayed. Error bars indicate standard errors. The online version of this article includes the following source data for figure 3:

**Source data 1.** Numerical data that are represented as a graph in *Figure 3*.

## Biased distribution of nonsense errors in RNA transcripts

As shown in *Table 1*, nonsense errors represent only a small percentage of all errors. However, they are of particular interest because they will result in the formation of a premature termination codon (PTC) and thus truncated proteins if not degraded. To ameliorate the potential severe fitness effects resulting from such errors, eukaryotes have evolved the Nonsense Mediated Decay (NMD) mechanism (*Losson and Lacroute, 1979*; *Maquat, 1995*; *Peltz et al., 1993*) to facilitate the degradation of RNA transcripts carrying PTCs. A key to the success of NMD is distinguishing a PTC from the original stop codon (*Amrani et al., 2004*; *Le Hir et al., 2001*), and the ability of the NMD machinery to identify a PTC is thought to diminish as the PTC approaches the 3′ end of a mRNA (*Isken and*

**Table 1.** Percentages of transcript errors in mRNAs that are synonymous, missense, or nonsense (other potential types of transcript errors with small percentages, such as start/stop codon loss-errors, are not shown).

Observed and expected (in parentheses) percentages are presented. Based on the bias of observed rNTP substitution rates and codon usages of each bacterium, expected percentages are calculated assuming a random generation of errors and an absence of error-correction processes. The level of significant difference is indicated by asterisks (*$P < 0.05$, ** $P < 0.005$, $\chi^2$ test).

| Species | Synonymous | Missense | Nonsense |
|---|---|---|---|
| *E. coli* | 40.18 (34.35) ** | 56.25 (59.79) * | 3.57 (5.62) ** |
| *B. subtilis* | 32.76 (31.86) | 61.69 (61.63) | 5.15 (6.15) |
| *A. tumefaciens* | 40.68 (36.76) * | 56.36 (59.17) | 2.96 (3.86) |
| *M. florum* | 17.58 (24.12) ** | 79.27 (70.58) ** | 2.37 (4.85) ** |

*Maquat, 2007*). This hypothesis is supported by yeast transcript-error data that show a marked increase in the frequency of PTCs towards the 3′ end of mRNAs (*Gout et al., 2017*).

Although no analog of the eukaryotic NMD system is known in prokaryotes, a destabilizing effect of PTCs on mRNA stability has been observed in bacteria (*Arnold et al., 1998*; *Braun, 1998*; *Morse and Yanofsky, 1969*; *Nilsson et al., 1987*). Evaluating the distribution of nonsense errors across the whole length of mRNA transcripts, we observed an increased frequency of nonsense errors at the 3′ end of transcripts, although the trend is not statistically significant in *A. tumefaciens* (*Figure 4A*). Compared to other three species, a smaller number of nonsense errors were detected in *A. tumefaciens* (*Supplementary file 1*, Table 7), which may result in a low statistical power to reveal a potential pattern for the distribution of nonsense errors. We further modified the analysis by dividing the frequency of nonsense errors by that of all errors. This ratio tends to be higher at the 3′ end of mRNAs (*Figure 4—figure supplement 1*), excluding the possibility that the enrichment of nonsense errors results mainly from a higher overall transcript-error rate at the 3′ end of mRNAs.

Of all types of genetic codons, those with one nucleotide difference from a stop codon (one-off codons) have a higher probability of mutating into PTCs. We further normalized the frequency of nonsense errors by the abundance of one-off codons at corresponding loci. This still revealed an increased frequency of nonsense errors towards 3′ ends of transcripts (*Figure 4—figure supplement 2*), suggesting the higher frequency of nonsense errors is not caused by more abundant one-off codons at the 3′ end of transcripts.

The increased frequency of PTCs at the 3′ end of mRNA transcripts suggests the presence of an NMD-like process, albeit by a likely different mechanism than in eukaryotes, which largely rely on the poly-A tail or exon-exon junction complex (*Amrani et al., 2004*). One speculative model for the degradation of PTCs in eukaryotes, the ribosome-release model (*Brogna and Wen, 2009*), in which the degradation of RNAs with PTCs depends on the degree of ribosome coverage on RNA molecules, has the potential to hold true in prokaryotes. Ribosomes can load on to nascent transcripts immediately after RNA synthesis. Therefore, a whole transcript with a normal stop codon can be covered by multiple ribosomes towards its 3′ end, with these ribosomes protecting the transcript from degradation by blocking ribonuclease cleavage sites. In contrast, a PTC upstream of the original stop codon will stall the ribosomes, leaving the ribonucleotides between the PTC and the site of the original stop codon unprotected by ribosomes, potentially promoting degradation by cellular ribonucleases (*Figure 4B*).

## Discussion

A key to accurately identifying *bona fide* transcript errors is to distinguish them from technical errors and low-frequency genetic mutations. With previous efforts on method development to eliminate sequencing errors (*Acevedo and Andino, 2014*; *Acevedo et al., 2014*; *Lou et al., 2013*) and to evaluate the error rate of the reverse transcriptase (*Gout et al., 2013*), it is now possible to ensure that contributions from such technical errors are orders of magnitudes lower than true transcript-error rates by the CirSeq approach (See Materials and methods). Except for *M. florum*, transcript-error rates in bacteria estimated by the current study are about one order of magnitude lower than those from a previous study (*Traverse and Ochman, 2016*). Specifically in *E. coli*, our error-rate estimates for each type of substitutions tend to be lower than those from *Traverse and Ochman (2016)*, the most striking difference involving the C→U substitution rate, which could be partly due to the use of a metal ion-based RNA fragmentation approach in the previous work vs. enzymatic RNA fragmentation in the present study. The latter minimizes RNA damage (*Gout et al., 2017*), in particular cytosine deaminations, introduced during the preparation of the sequencing library.

Besides base-substitution errors, a small portion of transcript errors can occur in other forms such as insertions and deletions. Estimates of transcript insertion/deletion (indel) error rates from species in this study are 0.1 to 0.2 of the corresponding base-substitution error rates (*Supplementary file 1*, Table 1).

Bacterial transcriptomes predominantly consist of ncRNA transcripts, such as rRNAs and tRNAs (*Westermann et al., 2012*). However, only a small portion of the whole ncRNA transcripts was evaluated in the present study (*Supplementary file 1*, Table 8) because of technical limitations. The rRNA depletion procedure in the sequencing library preparation protocol removes the majority of rRNAs. Secondary structures and nucleotide modifications of tRNAs interfere the cDNA synthesis and

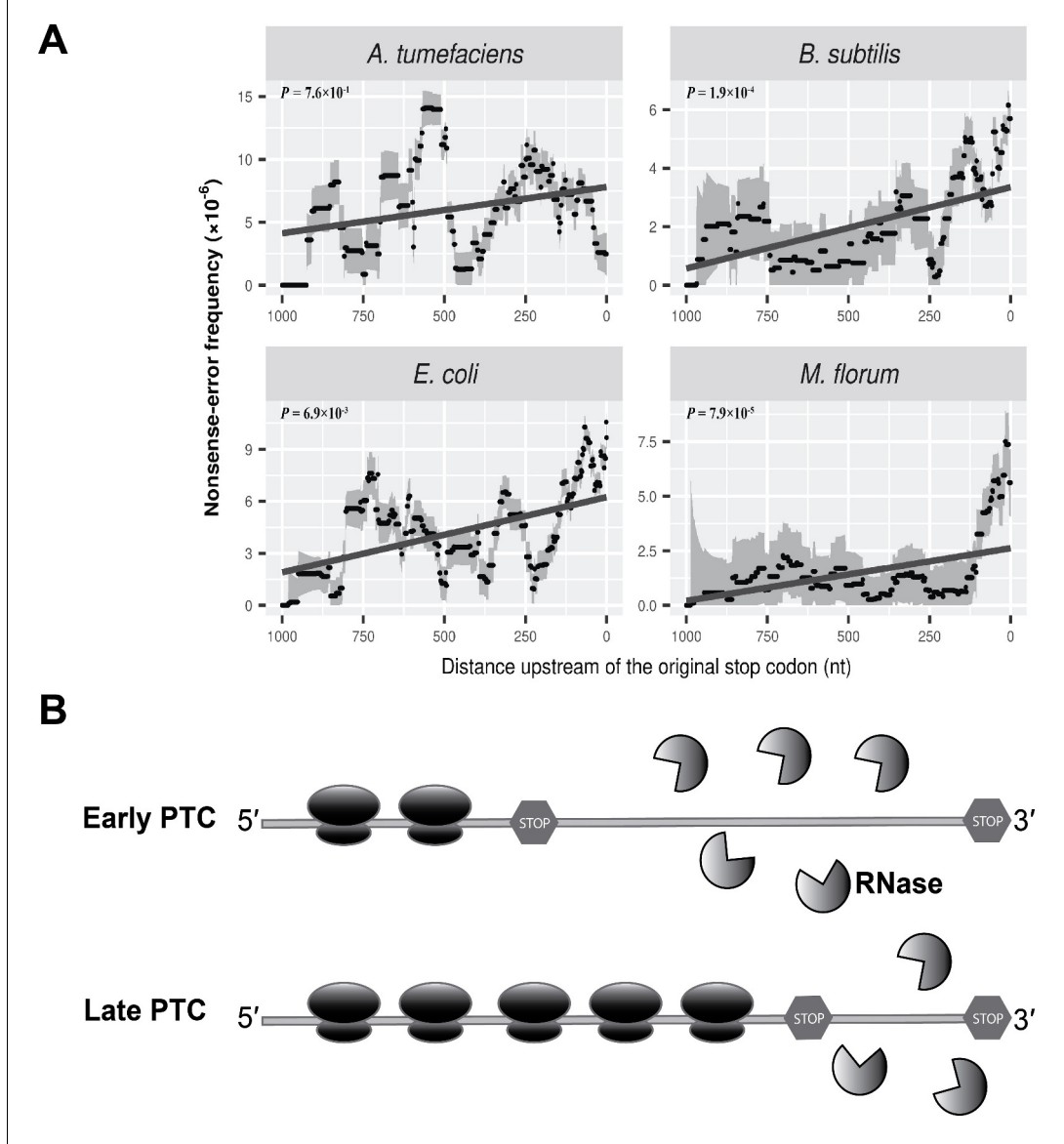

**Figure 4.** Nonsense errors in prokaryotic transcripts. (**A**) Distributions of nonsense errors across mRNA transcripts. The frequency of nonsense errors is calculated in a 100-nt sliding window with a step size of 1 nt for data visualization. Grey intervals represent standard deviations assuming the number of errors at each locus follows a binomial distribution. Linear regression between the distance to the original stop codon and the frequency of nonsense errors of each window is indicated in dark grey lines. *P* values were calculated from weighted linear regressions of individual data points before binning into a window. (**B**) The ribosome-release model for PTCs degradation in prokaryotes. Compared to a late PTC, an early PTC results in a larger portion of ribonucleotides unprotected by ribosomes, and therefore a higher probability of being digested by cellular ribonuclease.

The online version of this article includes the following source data and figure supplement(s) for figure 4:

**Source data 1.** Numerical data that are represented as a graph in *Figure 4A*.

**Figure supplement 1.** Distributions of the ratio of nonsense error frequency to total error frequency across mRNA transcripts.

**Figure supplement 1—source data 1.** Numerical data that are represented as a graph in *Figure 4—figure supplement 1*.

**Figure supplement 2.** Distribution of PTCs across the length of mRNA transcripts.

**Figure supplement 2—source data 1.** Numerical data that are represented as a graph in *Figure 4—figure supplement 2*.

sequencing adapter ligations. In the future, to achieve a better measurement of transcript-error rates of ncRNA transcripts, total RNAs can be mixed with rRNA-depleted RNAs at a certain ratio to increase the abundance of rRNAs in the sequencing library. Demethylase enzymes and thermophilic reversetranscriptase can be used to remove nucleotide modifications of tRNAs and to improve

processivity in generating cDNAs from highly structured RNA templates (*Schwartz et al., 2018*; *Zheng et al., 2015*).

The molecular spectrum of transcript errors revealed in our work indicates a general C→U substitution bias, which has been proposed to be due to spontaneous deamination (*Imashimizu et al., 2013*; *Traverse and Ochman, 2016*) owing to the chemical instability of cytosine (*Alberts et al., 2015*). Besides this widely accepted mechanism, non-Watson-Crick base pairing during rNTP incorporations may also contribute to this bias. Because dG and rU can form a base pair (*Sugimoto et al., 2000*; *Sugimoto et al., 1997*), mispairing between a template DNA (dG) and an RNA (rU) during rNTP incorporations likely also contributes to the C→U substitution bias.

Another intriguing observation from the molecular spectra in the present study is the G→A substitution bias in *M. florum*. One source for this substitution may be unrepaired uracils on the DNA antisense strand, which pair with rATPs during transcription, resulting in a G→A substitution on the RNA transcript. Although *M. florum* has a diminutive genome (0.79 Mb) and lacks many genes (RefSeq NC_006055.1), a uracil-DNA glycosylase (UDG) ortholog whose product presumably removes uracils (*McCullough et al., 1999*) does exist in the genome. Therefore, the extent to which mismatches between the unrepaired uracil and rATP can explain the G→A bias remains unclear.

Taking data from previous studies (*Gout et al., 2017*; *Imashimizu et al., 2015*; *Traverse and Ochman, 2018*) and this work together, G→A substitution bias seems to be a general pattern in cells with error-prone transcription machineries. What might be the underlying mechanism? The error spectrum is shaped by two factors. One is the ability of an RNA polymerase to distinguish correct rNTPs from incorrect ones. The other factor, which is sometimes neglected, is the rNTP pool within a cell. The error rate of competitive binding of rNTPs to the template can be expressed as, $(k_{incorrect} \cdot C_{incorrect-rNTPs})/(k_{correct} \cdot C_{correct-rNTPs})$, where $k$ refers to the rNTP incorporation rate and $C$ indicates the concentration of rNTPs. As suggested by this equation, a biased cellular rNTP concentration might present an additional challenge to transcriptional fidelity for certain categories of rNTPs. Based on observations that RNA polymerases have a low ability to distinguish rNTPs with the same structural class of nitrogenous bases and that the cellular concentration of rATPs is the highest among all types of nucleotides in both eukaryotes and prokaryotes (*Bennett et al., 2009*; *Buckstein et al., 2008*; *Traut, 1994*), it is reasonable to speculate that the high cellular concentration of rATPs contribute to the observed bias towards G→A substitutions.

An additional cellular process influencing transcript errors is RNA quality-control. Because genes involved in NMD, such as up-frameshift (UPF) genes, have not been identified in prokaryotes, evidence for the existence of NMD in prokaryotes is still lacking. However, previous studies based on single gene-reporters (*Baker and Mackie, 2003*; *Braun, 1998*; *Nilsson et al., 1987*) and our transcriptome-wide survey suggest a Nonsense-Mediated Decay-like quality-control mechanism in prokaryotes. A key implication of the increased frequency of nonsense errors at the 3′ end of mRNAs (*Figure 4A*) is that the degradation of RNAs carrying nonsense errors may simply result from a higher degree of exposure to cellular ribonucleases rather than from a reliance on specific protein-based systems.

Current models of mRNA surveillance mechanisms mostly focus on stop codon-related errors (*Deutscher, 2006*; *Richards et al., 2008*), which are expected to represent only a small portion of the total transcript errors in a cell. It is largely unknown whether, and if so by which mechanisms the major transcript errors (missense errors) get degraded. To resolve this, future research will be required to evaluate the rate at which transcript errors are degraded after initially being generated during transcription. This might be possible by comparing transcript errors on nascent transcripts bound to RNA polymerases with those on mature transcripts associated with ribosomes.

## Materials and methods

### Bacteria strains and growth conditions

All bacteria strains were inoculated into liquid culture from single colonies and grew to mid-exponential growth phase upon harvest. *E. coli* MG1655 and *B. subtilis* NCIB 3610 were grown at 37°C in LB liquid medium. *M. florum* L1 (ATCC #33453) was grown at 30°C in SNE liquid medium. *A. tumefaciens* C58 was grown at 28°C in LB liquid medium.

## RNA extraction

Bacteria were harvested from liquid culture media by centrifugation and total RNA was extracted and purified using the FastRNA Blue Kit (MPBiomedicals), RNase-free DNase set (Qiagen), and the RNeasy Mini Kit (Qiagen). rRNA was depleted by the Ribo-Zero rRNA Removal Kit (Bacteria) (Illumina) for the following library preparations.

## Library preparation and sequencing

We followed a refined protocol of CirSeq (*Gout et al., 2017*) to prepare libraries for transcript error identifications. Five hundred nanograms of rRNA-depleted RNAs were firstly fragmented with the NEBNext RNase III RNA Fragmentation Module (New England Biolabs) for 90 min at 37°C. After a clean-up using the Oligo Clean and Concentrator kit (Zymo Research), RNA fragments were circularized with RNA ligase 1 (New England Biolabs) according to the manufactuer's guidelines. cDNA with tandem repeats was generated by the rolling-circle reverse transcription as described in the refined CirSeq protocol. Synthesis of the second strand of cDNA and sequencing library preparation were performed using the NEBNext Ultra RNA Library Prep Kit and NEBNext Multiplex Oligos for Illumina (New England Biolabs). The size selection and clean-up during sequencing library preparations were performed by Agencourt AMPure XP Beads (Beckman Coulter) according the NEB guideline that is optimized for approximately 200nt RNA inserts. A final gel-based size selection was performed to enrich PCR amplified products that are longer than 300nt. Single-end reads (300nt) were then generated using Illumina HiSeq 2500 System. The sequencing data were deposited in NCBI with the Bio-Project Number PRJNA592142.

## Genome references and annotation files

The accession numbers of genome references for *E. coli*, *B. subtilis*, and *M. florum* are NC_000913.3, NZ_CM000488.1, and NC_006055.1. For *A. tumefacien*, accession numbers are NC_003062.2, NC_003063.2, NC_003064.2 and NC_003065.3. The corresponding genome annotation files are from RefSeq.

## Data analysis

Several analysis pipelines already existed to process reads with multiple tandem repeats and call transcript errors, but with their own limitations. The CirSeq_v2 pipeline (*Acevedo and Andino, 2014*; *Acevedo et al., 2014*) can only analyze reads with exactly three repeats and reads generated by CirSeq approach can contain more than four repeats if the original RNA template is smaller than 75 nt. Another pipeline described in a recent work in yeast (*Gout et al., 2017*) cannot generate consensus calls and recalculate the quality score from a site where not all base calls are identical. Therefore, we developed Python scripts following the methods outlined by *Lou et al. (2013)* (*Figure 1—figure supplement 2*). The structure of repeats within one read was identified by an autocorrelation-based method, in which the length of one potential repeat $P$ is detected by the maximum fraction of identical base calls that are separated by a distance $P$ within one read. The consensus sequence was constructed and the corresponding new quality score was calculated by a Bayesian approach where an inferred consensus call is taken with the maximum posterior probability given all observed base calls. This approach also allows the processing of varied numbers and types of base calls at one site. To identify the ligation junction of circular templates and to reorganize the consensus sequence, a tandem duplicate of the consensus sequence was constructed and then mapped back to the reference genome by BWA (*Li and Durbin, 2009*). The longest continuous mapped regions of the duplicated consensus sequences therefore correspond to original RNA fragments. We also excluded the 4 nucleotides at both ends of the reorganized consensus sequence to minimize potential confusions, because mapping can be ambiguous at the two ends of RNA fragments. After mapping of reconstructed consensus sequences, reads uniquely mapped to protein-coding regions and all reads mapped to ncRNA regions were kept. Transcript errors were called if a mismatch between a consensus call and the reference was supported by less than 1% of reads at corresponding loci. To exclude false positives of transcript errors from genetic mutations in multiple copies of ncRNA genes (such as rRNA and tRNA genes), an additional filter was included to exclude an error call that is supported by genetic variations from different copies of ncRNA genes. The transcript error rate of a given region was calculated as the number of transcript errors divided by the total number of rNTPs

assayed from the corresponding region. The code for the bioinformatic pipeline can be found at https://github.com/LynchLab/CirSeq4TranscriptErrors (*Li, 2020*; copy archived at https://github.com/elifesciences-publications/CirSeq4TranscriptErrors).

## Strategies to distinguish transcript errors from other types of errors

First, reverse transcription and sequencing errors need to be filtered out in the analysis. Because the rate of transcript error is generally $10^{-6} \sim 10^{-5}$ /nt , the recalculated probability of an erroneous base call at $10^{-7}$ or lower was required to minimize contaminations from sequencing errors. Because the error rate of the reverse transcriptase used here is $\sim 10^{-4}$ /nt (*Gout et al., 2013*), at least two tandem repeats were required in the analysis to minimize false positives from reverse transcription errors.

Second, genetic mutations (DNA level) can arise during cell culture and low frequency mutations can behave like transcript errors in the sequencing data. The probability of capturing a genetic mutation can be calculated by dividing the expected number of genetic mutations generated during cell propagations by the total transcriptome size at the time point of sample collection, $\frac{\mu \cdot g \cdot T \cdot n}{T \cdot n}$, in which $\mu$ is the per site per generation mutation rate, $g$ is the number of generations during cell culture, $T$ is the size of genome regions get transcribed, and $n$ is the average expression level per site. This equation can be further simplified as $\mu g$. Because we know the mutation rate from mutation accumulation experiments (*Lee et al., 2012*; *Lynch et al., 2016*; *Sung et al., 2016*; *Sung et al., 2015*; *Sung et al., 2012*) and the number of generations from culture-growth dynamics (~30 generations), Low frequency genetic mutation can only inflate the transcript-error rate we calculated here by ~1‰ -1%.

## To calculate the expected percentages of transcript errors with different effects

Take the calculation for synonymous substitution as one example. The percentage can be calculated by summing the probabilities of each codon to have a synonymous change, $P(syn) = \sum_{i=1}^{64} P_i \cdot P_{i(syn)}$. $P_i$ refers to the probability of having codon $i$ based on the codon usage of a specific genome and there are 64 codons in total. $P_{i(syn)}$ is the probability that codon $i$ has a synonymous substitution and it can be calculated from, $P_{i(syn)} = \sum_{j=1}^{9} \mu_j \cdot 1_{\{j\ results\ in\ syn\}}$. $\mu_j$ denotes the substitution probability of 1 of the 9 single-base substitutions that can happen in one codon. And it can be calculated by, $\mu_j = \frac{e_j}{\sum_{j=1}^{9} e_j}$, in which $e_j$ refers to the error rate of 1 of the 9 substitutions in one codon. Estimates of $e_j$ are displayed in *Figure 3*.

## The sliding window analysis and weighted linear regression to evaluate the distribution of nonsense errors on mRNA transcripts

The sliding window analysis (window size = 100nt and step size = 1nt) of the distribution of nonsense errors across mRNAs was used for data visualization. To evaluate whether or not the negative correlation between the frequency of nonsense errors and the corresponding distance from a nonsense error to the original stop codon is statistically significant, a weighted linear regression method was used. The weight was calculated as the reciprocal of a variance of a nonsense error frequency. Because the observed number of transcript errors at each locus is expected to follow a binomial distribution, the variance of the nonsense error frequency can be estimated as $\frac{p(1-p)}{n}$, where $p$ is the estimated frequency of errors and $n$ refers to the read coverage at the corresponding locus.

## Acknowledgements

We thank Stephen Simpson, W Kelley Thomas, Samuel F Miller, Jiaqi Zheng, Jie Huang, and James Ford for technical support, and Daniel Kearns and Clay Fuqua for providing *B. subtilis* NCIB 3610 and *A. tumefaciens* C58 strains. We also thank Hongan Long, Michelle Marasco, Chi-Chun Chen, Parul Johri, and Jean-Francois Gout for helpful discussions. This work was supported by National Institutes of Health Awards R01-GM036827 and R35-GM122566, and Multidisciplinary University

Research Initiative Award W911NF-09-1-0444 and W911NF-14-1-0411 from the US Army Research Office (to ML).

## Additional information

### Funding

| Funder | Grant reference number | Author |
|---|---|---|
| National Institutes of Health | R01-GM036827 | Michael Lynch |
| National Institutes of Health | R35-GM122566 | Michael Lynch |
| Army Research Office | W911NF-09-1-0444 | Michael Lynch |
| Army Research Office | W911NF-14-1-0411 | Michael Lynch |

The funders had no role in study design, data collection and interpretation, or the decision to submit the work for publication.

### Author contributions

Weiyi Li, Conceptualization, Resources, Data curation, Software, Formal analysis, Validation, Investigation, Visualization, Methodology, Writing - original draft, Writing - review and editing; Michael Lynch, Conceptualization, Supervision, Funding acquisition, Validation, Investigation, Writing - original draft, Project administration, Writing - review and editing

### Author ORCIDs

Weiyi Li https://orcid.org/0000-0002-1168-7093
Michael Lynch https://orcid.org/0000-0002-1653-0642

### Decision letter and Author response

Decision letter https://doi.org/10.7554/eLife.54898.sa1
Author response https://doi.org/10.7554/eLife.54898.sa2

## Additional files

### Supplementary files

• Supplementary file 1. Supplementary Table 1. Estimates of transcript indel error rates of four bacterial species. Standard error of the mean from three biological replicates are displayed. Supplementary Table 2—5. The Observed and expected numbers of transcript errors in genes of four bacterial species. Supplementary Table 6. The transition/transversion error bias revealed in four bacterial species. Supplementary Table 7. The observed and expected numbers/percentages of transcript errors that are synonymous, nonsynonymous, or nonsense. Supplementary Table 8. The numbers and rates of detected transcript errors that are from ncRNA transcripts in four bacterial species.

• Transparent reporting form

### Data availability

Sequencing data of this study are available at NCBI with the BioProject Number PRJNA592142.

The following dataset was generated:

| Author(s) | Year | Dataset title | Dataset URL | Database and Identifier |
|---|---|---|---|---|
| Weiyi Li, Michael L | 2020 | Transcript error studies on Escherichia coli, Bacillus subtilis, Agrobacterium tumefaciens, and Mesoplasma florum | http://www.ncbi.nlm.nih.gov/bioproject/?term=PRJNA592142 | NCBI BioProject, PRJNA592142 |

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
