## [Decision Letter]

**Acceptance summary:**

DNA replication errors are one of the engines of evolution. However, how errors accumulate at the different steps of the central dogma of molecular biology has received limited attention. Investigating how errors in transcription and translation affect cell biology first requires that we measure their rates and that we examine how they affect various species. In this study, the authors use a recently developed technology to measure transcription error rates in four species of bacteria. Their results reveal that transcription error rates are higher than mutation rates, that many of these errors lead to amino acid changes if translated and that there may be a mechanism in prokaryotes for the quality control of mRNAs. These results open new research avenues on how cells may control the propagation of errors along the steps of the central dogma and why such error rates may be tolerated by evolution.

**Decision letter after peer review:**

Thank you for submitting your work entitled "Universally high transcript error rates in bacteria" for consideration by *eLife*. Your article has been reviewed by a Senior Editor, a Reviewing Editor, and three reviewers. The following individuals involved in review of your submission have agreed to reveal their identity: Joanna Masel (Reviewer #3).

Our decision has been reached after consultation between the reviewers. Based on these discussions and the individual reviews below, we regret to inform you that your work will not be considered further for publication in *eLife*. But please keep in mind that a substantially changed version could be re-submitted as a new paper.

The reviewers agree that the questions addressed are interesting and important and consider that it is possible to improve the manuscript so that it meets their criticisms. They found particularly interesting the proposed mechanism based on premature termination but found that it was not fully supported by the analyses and may require actual lab experiments. They also identified several issues with the presentation of the data and some of the analyses. In light of this, we cannot consider the manuscript further under its present form. Their complete comments appear below.

Reviewer #1:

The manuscript submitted by Li and Lynch is interested in transcript error rates in bacteria, which were already known to be orders of magnitude more frequent than replication errors. The authors developed a slightly modified version of the previously published CirSeq approach, which produces tandem copies of transcribed fragments from circularized RNA molecules, to allow a more accurate detection of errors on a genome-wide scale. The authors used this method to quantify mutations in transcripts from four representative bacteria, and showed evidence suggesting that the M. florum RNA polymerase is error-prone, with a strong G to A substitution bias compared to the other bacteria. The authors showed data suggesting that nonsense mutations tend to accumulate at the 3' end of transcripts and proposed a model to explain how transcripts containing a premature termination codon could be recognized and degraded by the cell.

Overall, I found the topic interesting but the results rely on a single type of evidence (CirSeq) and are mostly descriptive. Additional experiments would be required in my opinion to fully support the model proposed by the authors. For example, it would be interesting to introduce a premature termination codon at different positions in a reporter gene such as lacZ and evaluate if such mutated transcripts have a shorter half-life compared to the wild-type. I am still debating whether this is essential for this manuscript or not. I also noted a few suggestions or elements that should be clarified:

- It would be useful for readers to include a more detailed description of CirSeq as the manuscript heavily depends on this technique. The manuscript was mentioning reference work but a short explanation would help.

- The average mutation rate of reverse transcriptase is ~1E-4. Since this rate is relatively high, the authors only considered mutations with at least two tandem repetitions in the CirSeq data. Were the reverse transcription errors observed by Gout et al., (2013) evenly distributed across the transcripts (hotspots), and were certain mutations types more predominant than others? If such biases exist, would it still be possible to confidently discriminate between reverse transcription errors versus transcription errors? Can the authors comment on that?

- In Figure 4 and Figure S1 to Figure S3, how were the replicates processed? Were they combined before the sliding window analysis or after? Would it be relevant to show the error frequency calculated for each replicate to see the inter-replicate variability? The Pearson's correlations coefficients should also be present on the figures, especially for Figure S1 as this comparison is very important for the conclusions. My first impression was that Pearson correlation coefficient from Figure S1 (all transcript errors) would be similar to those reported for Figure 4 (nonsense errors) but the comparison is not possible in this version of the manuscript. Is there a reason to not show the Pearson's correlation coefficient on Figure S1 but show them in Figure 4 and Figure S2- Figure S3? From what I understand in subsection “Biased distribution of nonsense errors in RNA transcripts”, the correlation coefficients calculated on data presented in Figure S1 should not be significant. Is that the case?

Reviewer #2:

The manuscript by Li and Lynch reports the mutational spectrum of transcription in 4 bacterial species. The objective, results and discussion are clear and easy to read. However, I found that the methodological descriptions often lack important technicalities that are needed (at least to me) to fully appreciate what the authors have done exactly. I will first raise few general points and then provide a list of missing methodological pieces of info. I have no expertise in the molecular biology part, so I won't comment on it.

- Subsection “A global view of the transcript error distribution”: If the authors used a Bonferroni correction for multiple testing, they should not observe 5% of false positives, but expect 0.05 false positives, that is well below 1. Typically they should observe 0 positive if H0 is correct. If they really observe 5%-ish (as it is implied in the text), they do have a lot of significant genes for which transcripts are enriched in mutations.

- Subsection “Biased distribution of nonsense errors in RNA transcripts”: the authors report an overrepresentation of non-sense mutations near the 3' of the transcript. This is interpreted as a sign of a potential NMD in bacteria. I indeed think this is indeed a possibility. Alternatively, it could also be the case that simply there are more mutations in the 3' end of transcript in general. What is the pattern for missense and synonymous mutations regarding their localization in the genes?

- Subsection “Biased distribution of nonsense errors in RNA transcripts”: I am not sure of what the added value of this measure of relative gene length. Can you get rid of this paragraph and plot?

- Did the authors attempted to look at indels? I think 'long enough' SSRs will also present SSRs copy number variations.

- for Abstract/Introduction: errors can also accumulate without replication nor transcription (see some recent papers on non-replicative/quiescent errors)

As mentioned earlier, many methodological pieces of information are missing. Here is a list of some of them. Consider that the readers must have *everything* to understand and eventually redo the experiments. In this current version, most treatments are opaque.

- Subsection “A global view of the transcript error distribution”: how did you normalized to "the same level"?

- Subsection “Characterization of transcript errors”: sure, ~2/3 of mutations are non-synonymous. So this does not come as a surprise.

- Subsection “Characterization of transcript errors”: "close to" means "close to significance"?

- Subsection “Data analysis”. Can you provide insights on what is the Lou et al., method? What is the autocorrelation method? Where is the python code available, so readers can have a chance to understand what you did? What is the Bayesian approach you mentioned using? What parameters of BWA did you used?

- Subsection “Strategies to distinguish transcript errors from other types of errors”: what is the equation? (you suggested simplifying by \mu g, but I don't see an equation)?

- Subsection “To calculate the expected percentages of transcript errors with different effects”: I suspect these are not probabilities but counts turned into frequencies, right? So at best, probabilities estimates. \mu_i are the relative mutation rates? More generally, I am not sure what did you use this whole calculation (in this paragraph) for?

- Figure 3 legend: what are 'conditional' error rates?

- Table1: How exactly did you compute your p-values? Did you take the spectrum into account?

- Figure 4: Why do you have the x-axis in reverse order? How did you normalize in 0-1.

In conclusion, I believe this manuscript has potential but should be revised with great care before becoming a decent published article.

*Reviewer #3*:This manuscript measures the single nucleotide transcription error rate and spectrum in four bacterial species. The primary findings are that *E. coli* errors are an order of magnitude less common than previously reported by Traverse and Ochman, that M. florum has a strikingly high G->A substitution bias, and that nonsense errors are depleted toward the 3' end of mRNA in a manner compatible with previously hypothesized NMD-like quality control in prokaryotes. Overall, a potential obstacle to publication in *ELife* is the incremental nature of these findings – the methods are not greatly advanced from earlier work including from the same group, this is not the first evidence for NMD-like quality control, and there is speculation about but not proof for the reasons for the high G->A bias and the discrepancy with previous *E. coli* measurements. I have a number of concerns, especially about the statistics, but correcting them is unlikely to reverse the major findings.

Probably the biggest issue is the contrast between Figure 4 and Figure S3, used to help infer the mechanism of action of the NMD-like system. Frequencies are a kind of binned data, and it is inappropriate to report correlation coefficients from binned data: see eg https://statmodeling.stat.columbia.edu/2016/06/17/29400/ or https://serialmentor.com/blog/2013/8/18/common-errors-in-statistical-analyses for discussions of this point. It is definitely not appropriate to claim that the Figure 4 model is better than the Figure S3 model (subsection “Biased distribution of nonsense errors in RNA transcripts”) because the correlation coefficients are larger, especially because the binning is clearly different in the two cases, eg. with far more zeros in Figure S3.

For these and all similar figures, there should be error bars on each dot from sampling error: for a binomial the error is sqrt(p*(1-p)/n). I was unable to figure out what exactly the normalization to scale the y-axis between 0 to 1 entailed – the fact that no code was made available (despite the *ELife* reporting form instructions to "Include code used for data analysis") meant that I couldn't compensate for thinly described methods. I see in any case no justification for normalization; it would be better to just give the actual numbers on the y-axis, while keeping the "zoom" the same for visualization purposes.

The best statistical approach is generally to work with the raw data, which in this case is a vast dataset of 0s (no error) and 1s (error). The number of (rare) errors meeting given criteria follows a Poisson distribution, and a generalized linear model can be used to model this error function. This is what we did for transcription errors in https://www.biorxiv.org/content/10.1101/554329v1, and it leads to vastly greater power to discern the kinds of trends hypothesized in Figure 4 and Figure S3. I realize that, as well as some conflict in asking for use and thus citation of my own work, there may be concerns citing a preprint, but the manuscript is now accepted pending minor revisions in GBE, and citable as such. I would really like to know eg whether M. florum really does have a different shape to *E. coli* in Figure 4 as it appears to, and more sophisticated statistical models are required to answer questions of this sort. Our accepted manuscript provides such models, with code fully available on github.

Our preprint re-analyzes the data of Traverse and Ochman, and finds that the (non-C->U) error rate depends on protein abundance. The Discussion section is the only place in the paper that attempts to say why the currently observed non-C->U error rate is an order of magnitude lower than that previously observed by Traverse and Ochman, attributing all the discrepancy, especially but not limited to cytosine deamination, to RNA damage during library preparation by Traverse and Ochman. This is problematic in the light of our finding of systematic differences that are not expected to be caused by library preparation problems. I don't know why the non-C->U error rates are so different between the studies either, but it clearly isn't all cytosine deamination, nor all library preparation, and it would be useful to acknowledge this puzzle and comment on other possibilities, e.g. differences in strain or experimental condition.

Note that the per-gene method to find outliers described in subsection “A global view of the transcript error distribution” is much lower powered than our test for dependence on protein abundance, and so in no way rules out the variation among genes that we discovered. And if the aim is to detect mutations or programmed errors, it is better to do so per-site than per-gene, as we also did to also find that such problems were rare.

In general, the statistics used in this manuscript assume that sampling error is negligible and that all variance is therefore attributed to biological replication. This underlies the use of t-tests and paired t-tests throughout. However, in tables like Supplementary file 5 and Supplementary file 6, neither error bars nor denominator is given, so I am unable to verify that sqrt(p*(1-p)/n) is negligible. If it is not negligible, then all these t-tests should be weighted rather than the current use of unweighted tests, or better still, a more advanced generalized linear model with a Poisson error term (see above) that simultaneously accounts for all sources of variance should be used instead of t-tests. The problems with the statistical tests used primarily create lower power, and so the positive findings should all hold, but more problematic is that the supplementary data files lack the information needed to allow future reanalysis of the data using better methods.

Another statistical problem is in calculating expected percentages (subsection “To calculate the expected percentages of transcript errors with different effects”). The equation used assumes that while the error rates of the 3 sites within the codon might be different, every codon has an equal error rate. The much more logical alternative would be to assume site-specific error rates that do not depend on which codon the site is found in. If our finding that error rates depend on protein abundance is also true, this will contribute to misleading results even after this correction is made. We found that the difference between error rates at synonymous vs. non-synonymous sites was entirely attributable to the dependence of codon bias strength on protein abundance, i.e. that variation in error rate was at the gene level not the codon level.

Figure 2 legend refers to the analysis only of genes with detected transcript errors. This would seem to create a variety of ascertainment biases, inflating the error rate overall as zeros are neglected, and preferentially neglecting potentially large numbers of genes each of low expression.

[Editors’ note: further revisions were suggested prior to acceptance, as described below.]

Thank you for submitting your article "Universally high transcript error rates in bacteria" for consideration by *eLife*. Your article has been reviewed by Patricia Wittkopp as the Senior Editor, a Reviewing Editor, and three reviewers. The following individuals involved in review of your submission have agreed to reveal their identity: Joanna Masel (Reviewer #2).

The reviewers have discussed the reviews with one another and the Reviewing Editor has drafted this decision to help you prepare a revised submission.

Summary:

In this manuscript, Li and Lynch measure the rate of transcription errors in a few prokaryotes. This is a revised version of a previous manuscript. The reviewers appreciate the importance of the work and the changes that have been made to the initial version. However, there are still some concerns about the statistical analyses and interpretation of the data. The comments are summarized below.

Essential revisions:

Data analysis and interpretation:

1) They uncovered almost the full spectrum of transcription errors, but somehow omitting indel errors from their data set. As is, this work will serve as a method reference for people interested in transcription errors. Based on their spectrum analysis, the authors speculate about different mechanisms by which these errors can be made. In order to be complete, the authors should analyze and report indels from their existing data.

2) Concerning the model to explain the biased distribution of nonsense errors in mRNA, they proposed an NMD mechanism no yet characterized in bacteria, but this can easily be explained by transcription polarity described first by Morse and Yanofsky, (1969).

3) In order to make the sequencing method user-friendly, I would strongly suggest that the authors include a flowchart for the analysis of the sequencing data as they have done for the technical concept in Figure 1—figure supplement 1.

Statistical analyses:

4) While claims based on comparing the magnitudes of correlation coefficients of different binned data sets has now been removed, correlation coefficients for data that is not just binned (frequency data), but also non-independent (sliding windows) continue to be reported in Figure 4, Figure S2, and Figure S3. Corresponding regression models on these non-independent sliding window datapoints are inappropriately used to generate p-values. An alternative and somewhat less flawed approach is given only in the response to reviews document and not in the manuscript itself. This is presented as being the method of Meer et al., "with minor modifications", although they are not that minor, as they sacrifice the main advantage of that method. A good statistical model should have a good model of the error function. Count data has a binomial error function, well approximated as a Poisson. The advantage of the method of Meer et al., is that it exactly models this Poisson, but the method is dismissed in the response to reviews because it explicitly accounts for the fact that the raw count data depends on the number of reads, as though this were a bad thing. The alternative presented in the response to reviews does avoid the non-independence problem of the sliding window, but it still assumes homoscedasticity with respect to #errors/#reads, an assumption that is violated in practice given many more reads at some sites than others. If the authors want to model Equation (1) instead of Equation (2) (as numbered in the response to reviews), I would find this acceptable if and only if they (a) use a weighted regression to account for the known heteroscedasticity (ie at least model the magnitude of the error even if they don't model its shape), and (b) place this analysis in the manuscript itself as a replacement for doing a regression on the output of non-independent sliding-window-bins. The sliding window is perfectly appropriate and indeed very helpful as a visualization of the data, but inappropriate for the fitting and testing of statistical models.

5) I take issue with the claim in subsection “A global view of the transcript error distribution” that transcript errors are randomly distributed across genes. After a Bonferroni correction for the number of genes, the manuscript has very little power with which to make such a claim of evidence of absence. What is more, Meer et al., showed (using a different *E. coli* dataset) that the error rate does depend on the strength of selection on the gene in question, as assessed by its protein abundance. This analysis had much higher power than the current study, firstly because there was only one degree of freedom and hence no need for a radical Bonferroni correction, and secondly because explicit modeling of the error function as described above will generally increase power. Given that (uncited) evidence that contradicts the manuscript's claim for randomness has already been published, and the lower power of the current manuscript's basis for making this claim, a claim of randomness should not be made unless the authors are able to apply the same or similarly high-powered method and directly show contradictory results. Should the previously published claim non-randomness be supported, this would have implications for the authors' model of randomly generated transcript error, eg subsection “Characterization of transcript errors”. In this passage too, the statistics are problematic. Observed and expected "data" are compared using a paired t-test, as though the expectations were themselves data with error magnitudes homoscedastic with those of the observed data.

6) I am also confused by the bootstrapped standard deviation in Figure 4, Figure S2, and Figure S3. In Figure 4, why bootstrap rather than use the known formula for a binomial? In Figure S2 and Figure S3, why bootstrap only the numerator and not the denominator of the quantity whose error is being assessed?

7) I still fear that the casual reader will take away from the Discussion section that differences between the two studies in error rates other than C->U are trivial in magnitude, which is not the case.

---

## [Author Response]

In this revision, we have addressed all the reviewers’ comments. In response to the major comment and to better support the Nonsense Mediated Decay-like mechanism in prokaryotes, refined analysis methods and statistical tests have been provided to evaluate the distribution of nonsense errors across mRNAs. Extensive empirical evidence has also been cited to support the destabilizing effect of nonsense errors to prokaryotic mRNAs.

Reviewer #1:[…] Overall, I found the topic interesting but the results rely on a single type of evidence (CirSeq) and are mostly descriptive. Additional experiments would be required in my opinion to fully support the model proposed by the authors. For example, it would be interesting to introduce a premature termination codon at different positions in a reporter gene such as lacZ and evaluate if such mutated transcripts have a shorter half-life compared to the wild-type. I am still debating whether this is essential for this manuscript or not. I also noted a few suggestions or elements that should be clarified:

It would certainly be interesting to do this experiment, but one would also have to be cautious to interpret results from these reporter-construct assays because there may be gene-specific peculiarities. In contrast, nonsense errors detected by CirSeq are from a transcriptome-wide study. It is hard to see how CirSeq could cause the bias observed in Figure 4.

However, experiments similar to that suggested by the reviewer have actually been done previously, with nonsense mutations introduced to different positions of reporters, such as bla, rpsO, and rpsT genes. The dynamics of mRNA degradation of these mutants are fully consistent with our results, therefore providing further support for the ribosome-release model. Please see:

Figure 2A of https://www.ncbi.nlm.nih.gov/pubmed/2440033

Figure 3 of https://www.ncbi.nlm.nih.gov/pubmed/9707438

Figure 3CD, Table 1. Entry 14, 15 of https://onlinelibrary.wiley.com/doi/full/10.1046/j.1365-2958.2003.03292.x

All these results are now cited in the Discussion of the manuscript.

- It would be useful for readers to include a more detailed description of CirSeq as the manuscript heavily depends on this technique. The manuscript was mentioning reference work but a short explanation would help.

Thanks for this suggestion. We have included a short description on crucial steps in making the sequencing library. We also include a flowchart as Figure 1—figure supplement 1 to illustrate how this approach provides an accurate detection of transcript errors.

- The average mutation rate of reverse transcriptase is ~1E-4. Since this rate is relatively high, the authors only considered mutations with at least two tandem repetitions in the CirSeq data. Were the reverse transcription errors observed by Gout et al., (2013) evenly distributed across the transcripts (hotspots), and were certain mutations types more predominant than others? If such biases exist, would it still be possible to confidently discriminate between reverse transcription errors versus transcription errors? Can the authors comment on that?

Indeed, as shown in Figure 1—figure supplement 1 of Gout et al., (2013), some types of reverse transcription (RT) errors are more predominant than others. However, the highest RT error rate is < 1.5x10^-4^, so the probability of observing the same RT error in two tandem repeats and misidentifying it as a transcript error is < 2.25x10^-8^. Because the transcript-error rates that we observe are on the order of 10^-6^ to 10^-5^, RT errors cannot contribute more than ~2% of detected transcript errors by including at least two tandem repeats of CirSeq reads.

In summary, whereas in principle it might be possible to finesse a maximum-likelihood procedure to factor in RT errors in a full data set, this would be error-prone itself, only introducing another source of sampling variance in the final analysis, and given the points made above, we are confident that RT errors are not contributing to our estimates in more than a very minor way.

-In Figure 4 and Figure S1 to Figure S3, how were the replicates processed? Were they combined before the sliding window analysis or after? Would it be relevant to show the error frequency calculated for each replicate to see the inter-replicate variability? The Pearson's correlations coefficients should also be present on the figures, especially for Figure S1 as this comparison is very important for the conclusions. My first impression was that Pearson correlation coefficient from Figure S1 (all transcript errors) would be similar to those reported for Figure 4 (nonsense errors) but the comparison is not possible in this version of the manuscript. Is there a reason to not show the Pearson's correlation coefficient on Figure S1 but show them in Figure 4 and Figure S2- Figure S3? From what I understand in subsection “Biased distribution of nonsense errors in RNA transcripts”, the correlation coefficients calculated on data presented in Figure S1 should not be significant. Is that the case?

We apologize for the confusion. In the sliding-window analysis as shown in Figure 4, we first combined the data from three biological replicates for one species because the number of nonsense errors detected from each replicate is small.

In our previous study in yeast (https://advances.sciencemag.org/content/3/10/e1701484, Figure 4F), we found that the expected pattern (enrichment of PTCs at 3’ end of RNAs) is only observable with a large number of nonsense errors. (The pattern is not shown in WT but shown in transcriptional fidelity mutants).

The number of nonsense errors of each biological replicate ranges from 10 to 50, which is small.

Therefore, we merged the data from biological replicates before the sliding-window analysis to have better power to reveal the overall pattern. We have also included statistics such as regression coefficients and sampling errors in Figure 4.

Figure S1 has now been updated as Figure S2, in which the ratio of nonsense error frequency to total error frequency is plotted. The higher ratio at the 3’ end excludes the possibility that the enrichment of nonsense errors results mainly from a higher overall transcript-error rate at the 3′ end of mRNAs.

Reviewer #2:The manuscript by Li and Lynch reports the mutational spectrum of transcription in 4 bacterial species. The objective, results and discussion are clear and easy to read. However, I found that the methodological descriptions often lack important technicalities that are needed (at least to me) to fully appreciate what the authors have done exactly. I will first raise few general points and then provide a list of missing methodological pieces of info. I have no expertise in the molecular biology part, so I won't comment on it.

To minimize the confusion on the refined CirSeq method, we have included more descriptions and added a flowchart as Figure 1—figure supplement 1 to clarify this approach.

- Subsection “A global view of the transcript error distribution”: If the authors used a Bonferroni correction for multiple testing, they should not observe 5% of false positives, but expect 0.05 false positives, that is well below 1. Typically they should observe 0 positive if H0 is correct. If they really observe 5%-ish (as it is implied in the text), they do have a lot of significant genes for which transcripts are enriched in mutations.

We have modified the analysis, first identifying genes enriched with transcript errors from each biological replicate. We define a gene as a hotspot if it is supported by at least two biological replicates. As shown in Supplementary file 1, Supplementary file 2, Supplementary file 3 and Supplementary file 4, there are just a few hotspot genes (ranging from 0-4 in each species), and transcript errors are in general randomly distributed across genes.

- Subsection “Biased distribution of nonsense errors in RNA transcripts”: the authors report an overrepresentation of non-sense mutations near the 3' of the transcript. This is interpreted as a sign of a potential NMD in bacteria. I indeed think this is indeed a possibility. Alternatively, it could also be the case that simply there are more mutations in the 3' end of transcript in general. What is the pattern for missense and synonymous mutations regarding their localization in the genes?

To exclude the alternative hypothesis proposed here, we have further divided the window-specific PTC frequencies by the overall error frequencies and evaluated the distribution of this resultant value. Figure S2 has been updated with this result and suggests that the enrichment of PTCs at the 3’ end of RNAs cannot be simply explained by more transcript errors at 3’ end than other regions.

- Subsection “Biased distribution of nonsense errors in RNA transcripts”: I am not sure of what the added value of this measure of relative gene length. Can you get rid of this paragraph and plot?

We did this analysis because we intended to test whether the efficiency of degradation depends on the absolute size or the portion of the ribonucleotides of one transcript that are uncovered by ribosomes.

As we revised the manuscript, we noticed that most bins in Figure S3 of the last version contained zeros and the sample size of nonsense errors in the present study does not allow us to compare the above hypotheses. Therefore, we have removed Figure S3 and the corresponding paragraph.

- Did the authors attempted to look at indels? I think 'long enough' SSRs will also present SSRs copy number variations.

We haven’t evaluated indel errors because an algorithm that is accurate and efficient to reconstruct consensus sequences from tandem repeats with potential indels is still not available. We agree with the reviewer that copy number variations of SSRs may happen at the transcriptome level, and that this piece of information may shed light on the mechanism of the slippage of RNA polymerases. More extensive analysis on indels will be a future direction of research.

- for Abstract/Introduction: errors can also accumulate without replication nor transcription (see some recent papers on non-replicative/quiescent errors)

We agree. Genetic mutations can be generated by non-replicative mechanisms such as DNA damage and DNA repair errors. We did not intend to review all mechanisms of errors, and our only point here is to draw attention from replication to transcription.

As mentioned earlier, many methodological pieces of information are missing. Here is a list of some of them. Consider that the readers must have *everything* to understand and eventually redo the experiments. In this current version, most treatments are opaque.- Subsection “A global view of the transcript error distribution”: how did you normalized to "the same level"?

We have updated details on how transcript-error rates were calculated in the legend of Figure 2.

- Subsection “Characterization of transcript errors”: sure, ~2/3 of mutations are non-synonymous. So this does not come as a surprise.

This piece of result is not surprising, but essential to provide a full picture of the potential functional categories of transcript errors.

- Subsection “Characterization of transcript errors”: "close to" means "close to significance"?

“close to” means the observed values are close to random expectations.

- Subsection “Data analysis”. Can you provide insights on what is the Lou et al., method? What is the autocorrelation method? Where is the python code available, so readers can have a chance to understand what you did? What is the Bayesian approach you mentioned using? What parameters of BWA did you used?

The autocorrelation method meant to identify the structure of tandem repeats and the Bayesian approach to recalculate quality scores of consensus reads are proposed in Lou et al. We have posted these python scripts on Lynch-lab Github site.

The BWA cmd that we used is the following:

bwa mem reference sample.fastq > sample.sam

There are multiple filters, so as to keep uniquely mapped reads and to keep high quality reads, after this BWA mapping step. We also posted this analysis pipeline on Github.

- Subsection “Strategies to distinguish transcript errors from other types of errors”: What is the equation? (you suggested simplifying by \mu g, but I don't see an equation)?

We apologize for the confusion. We have updated with the original equation and explained how to derive the simplified equation.

- Subsection “To calculate the expected percentages of transcript errors with different effects”: I suspect these are not probabilities but counts turned into frequencies, right? So at best, probabilities estimates. \mu_i are the relative mutation rates? More generally, I am not sure what did you use this whole calculation (in this paragraph) for?

The reviewer is correct. Pi is the estimate of probabilities based on frequencies. The μj can be understood as relative substitution rates of each of the 9 base substitutions in a codon. Equations here are used to calculate the expected percentages shown in Table. 1.

- Figure 3 legend: what are 'conditional' error rates?

Conditional error rates refer to error rates based on four different kinds of rNTPs. The way to calculate the conditional error rates has been explained in the legend of Figure 3.

- Table1: How exactly did you compute your p-values? Did you take the spectrum into account?

Yes, we have considered the bias in the molecular spectra of transcript errors. The equation to calculate the expected percentage can be found in the Materials and methods section. *P* values are from paired t-tests. Details are shown in Supplementary file 6.

- Figure4: Why do you have the x-axis in reverse order? How did you normalize in 0-1.

We have the x-axis in the reverse order to make it in a 5’ to 3’ orientation.

In the previous version of Figure 4, first we calculated PTC frequencies of corresponding sliding windows. Second, we simply rescaled these values from 0-1 by (value-min)/(max-min). However, to make the y-axis more straightforward, we don’t do the rescale now.

In conclusion, I believe this manuscript has potential but should be revised with great care before becoming a decent published article.Reviewer #3:This manuscript measures the single nucleotide transcription error rate and spectrum in four bacterial species. The primary findings are that *E. coli* errors are an order of magnitude less common than previously reported by Traverse and Ochman, that M. florum has a strikingly high G->A substitution bias, and that nonsense errors are depleted toward the 3' end of mRNA in a manner compatible with previously hypothesized NMD-like quality control in prokaryotes. Overall, a potential obstacle to publication in ELife is the incremental nature of these findings – the methods are not greatly advanced from earlier work including from the same group, this is not the first evidence for NMD-like quality control, and there is speculation about but not proof for the reasons for the high G->A bias and the discrepancy with previous *E. coli* measurements. I have a number of concerns, especially about the statistics, but correcting them is unlikely to reverse the major findings.Probably the biggest issue is the contrast between Figure 4 and Figure S3, used to help infer the mechanism of action of the NMD-like system. Frequencies are a kind of binned data, and it is inappropriate to report correlation coefficients from binned data: see eg https://statmodeling.stat.columbia.edu/2016/06/17/29400/ or https://serialmentor.com/blog/2013/8/18/common-errors-in-statistical-analyses for discussions of this point. It is definitely not appropriate to claim that the Figure 4 model is better than the Figure S3 model (subsection “Biased distribution of nonsense errors in RNA transcripts”) because the correlation coefficients are larger, especially because the binning is clearly different in the two cases, eg. with far more zeros in Figure S3.

We agree with the reviewer that the correlation coefficient from binned data should be avoided in evaluating the correlation between the frequency of nonsense errors and the distance between the nonsense error and the position of the original stop codon in a single base-resolution.

We did the sliding-window (99% overlap of neighboring windows) analysis in the present study for the following reasons:

1) Considering the sample size of nonsense errors (n=59-108) in each species, we can’t evaluate the distribution of the frequency of nonsense errors at single-base resolution. Our goal is simply to compare the frequency of nonsense errors at the very 3’ end to that at more upstream regions.

2) This being said, we have evaluated the correlations using individual points before binning (details shown in the following response), and the results are consistent with the sliding-window analysis.

3)Instead of binning data into separated windows, the step size of our sliding-window analysis is 1 nt and there is a 99% overlap between neighboring windows. This approach allows us to smooth fluctuations at a small scale, considering the small sample size of nonsense errors.

We also agree that it is not appropriate to claim a lack of correlation in analyses as shown in the Figure S3 because there are a lot of zeros. Data collected in the present study cannot help us to distinguish whether the absolute or relative distance between PTCs and 3’ end of RNAs matters for the efficiency of degradation. Therefore, we have removed the Figure S3 of the previous version and corresponding paragraphs.

For these and all similar figures, there should be error bars on each dot from sampling error: for a binomial the error is sqrt(p*(1-p)/n). I was unable to figure out what exactly the normalization to scale the y-axis between 0 to 1 entailed – the fact that no code was made available (despite the ELife reporting form instructions to "Include code used for data analysis") meant that I couldn't compensate for thinly described methods. I see in any case no justification for normalization; it would be better to just give the actual numbers on the y-axis, while keeping the "zoom" the same for visualization purposes.

We have included more details on how we ran analyses related to Figure 4, and error bars are included. We have also posted scripts and corresponding data on the Lynch-lab Github site.

The best statistical approach is generally to work with the raw data, which in this case is a vast dataset of 0s (no error) and 1s (error). The number of (rare) errors meeting given criteria follows a Poisson distribution, and a generalized linear model can be used to model this error function. This is what we did for transcription errors in https://www.biorxiv.org/content/10.1101/554329v1, and it leads to vastly greater power to discern the kinds of trends hypothesized in Figure 4 and Figure S3. I realize that, as well as some conflict in asking for use and thus citation of my own work, there may be concerns citing a preprint, but the manuscript is now accepted pending minor revisions in GBE, and citable as such. I would really like to know eg whether M. florum really does have a different shape to *E. coli* in Figure 4 as it appears to, and more sophisticated statistical models are required to answer questions of this sort. Our accepted manuscript provides such models, with code fully available on github.

We followed the approach suggested by the reviewer with minor modifications and got consistent results with our sliding-window analysis. Details are as follows.

The frequency of nonsense errors can be modeled as a linear function of the distance between nonsense errors and original stop codons,(1)EiRi=α+βdiwhere Ei is the number of nonsense errors at locus i, Ri refers to the number of ribonucleotides assayed, and di is the distance. To better evaluate this linear model, the reviewer suggested to multiply Ri on both sides and to evaluate the following model,(2)Ei∼Ri+Ri•di+εpoisson

However, the coefficient of the second term (Ri•di) in function (2) is not only related to di.

Therefore, we evaluated the linear model shown in (1) using the data points without binning.

Slopes (β) are -3.43E-07, -1.20E-07, -2.31E-07, and -1.32E-07 for *E. coli, A. tumefaciens, B. subtilis* and *M. florum*, respectively with *P* values of 6.01E-09, 4.02E-10, 0.019, and 7.63E-09. These results still support our argument that there is an enrichment of PTCs at the 3’ end (Figure 4A, in which the x axis is reversed).

Our preprint re-analyzes the data of Traverse and Ochman, and finds that the (non-C->U) error rate depends on protein abundance. The Discussion section is the only place in the paper that attempts to say why the currently observed non-C->U error rate is an order of magnitude lower than that previously observed by Traverse and Ochman, attributing all the discrepancy, especially but not limited to cytosine deamination, to RNA damage during library preparation by Traverse and Ochman. This is problematic in the light of our finding of systematic differences that are not expected to be caused by library preparation problems. I don't know why the non-C->U error rates are so different between the studies either, but it clearly isn't all cytosine deamination, nor all library preparation, and it would be useful to acknowledge this puzzle and comment on other possibilities, e.g. differences in strain or experimental condition.

The most striking difference comes from the C-to-U error rate. As we discussed in the manuscript, a lower C-to-U error rate in our study may result from enzymatic fragmentation approach which minimizes the RNA damages, in particular cytosine deaminations.

We agree with the reviewer that non C-to-U error rates are also different between the present and Traverse and Ochman’s studies, and note that it is entirely possible that metal ion-based fragmentation may also induce non C-to-U error rates.

We have provided details on our data and analysis methods, such as the strain number, growth phase, and criteria to filter false positives, which may be helpful for future studies to compare with our work. We realize that the reviewer’s recent conclusion in a theoretical paper relies on the previous Traverse data set, which we believe has significant issues, but it is not clear that this should bear on a judgement of our empirical work.

Note that the per-gene method to find outliers described in subsection “A global view of the transcript error distribution” is much lower powered than our test for dependence on protein abundance, and so in no way rules out the variation among genes that we discovered. And if the aim is to detect mutations or programmed errors, it is better to do so per-site than per-gene, as we also did to also find that such problems were rare.

We followed the per-site method described in the paper that the reviewer suggested. For example, in an *E. coli* sample, we calculated the probability that the transcript-error frequency at one site is higher than the random expectation according to cumulative binomial distribution. A significance cutoff of 5.4x10-9 (Bonferroni corrected p-value of 0.02) was used. A hotspot site was defined as being supported by at least two biological replicates. Results are shown below (Author response image 1).

**Author response image 1. sa2fig1:** 

Running this per-site method, first, we found transcript errors are generally randomly distributed, which is consistent with the per-gene method. Second, we also found the bias in different base substitution rates needs to be considered in this per-site model. For example, in *M. florum*, where the most hotspots were inferred, most hotspot sites correspond to GA errors. If a GA specific error rate is used, the number of hotspot sites decreases to 10. Lastly, we found that the feasibility of the per-site method depends on a high coverage at each site, which is not always the case in practice.We emphasize that all of our data will be freely available to the reviewer for further analysis.

In general, the statistics used in this manuscript assume that sampling error is negligible and that all variance is therefore attributed to biological replication. This underlies the use of t-tests and paired t-tests throughout. However, in tables like Supplementary file 5 and Supplementary file 6, neither error bars nor denominator is given, so I am unable to verify that sqrt(p*(1-p)/n) is negligible. If it is not negligible, then all these t-tests should be weighted rather than the current use of unweighted tests, or better still, a more advanced generalized linear model with a Poisson error term (see above) that simultaneously accounts for all sources of variance should be used instead of t-tests. The problems with the statistical tests used primarily create lower power, and so the positive findings should all hold, but more problematic is that the supplementary data files lack the information needed to allow future reanalysis of the data using better methods.

To provide information for future reanalysis, we have included details, such as numerators and denominators in tables like Supplementary file 5 and Supplementary file 6.

Another statistical problem is in calculating expected percentages (subsection “To calculate the expected percentages of transcript errors with different effects”). The equation used assumes that while the error rates of the 3 sites within the codon might be different, every codon has an equal error rate. The much more logical alternative would be to assume site-specific error rates that do not depend on which codon the site is found in. If our finding that error rates depend on protein abundance is also true, this will contribute to misleading results even after this correction is made. We found that the difference between error rates at synonymous vs. non-synonymous sites was entirely attributable to the dependence of codon bias strength on protein abundance, i.e. that variation in error rate was at the gene level not the codon level.

First, we would like to clarify the method we used to calculate the expected percentages. There are two biases we considered. The first is the bias in codon usages. The second is the bias in nucleotide-specific error rates. We do not assume every codon has an equal error rate; instead we calculated the error rate of each type of codon based on the nucleotide-specific error rate.

Second, we would like to argue that the finding from the reviewer’s study that the variation in error rate is at the gene level is only supported by the *E. coli* data set from Traverse and Ochman. It is unclear whether it holds true for other three bacterial species in the present study.

Lastly, the reviewer suggested to use site/gene-specific error rates to approximate empirical observations. However, we intend to compare expected percentages calculated from random expectations to observed percentages to reveal potential error-correction processes.

Figure 2 legend refers to the analysis only of genes with detected transcript errors. This would seem to create a variety of ascertainment biases, inflating the error rate overall as zeros are neglected, and preferentially neglecting potentially large numbers of genes each of low expression.

We apologize for the confusing wording here. We included all genes with adequate sequencing coverage, no matter whether there were errors detected or not.

[Editors’ note: what follows is the authors’ response to the second round of review.]

Essential revisions:Data analysis and interpretation:1) They uncovered almost the full spectrum of transcription errors, but somehow omitting indel errors from their data set. As is, this work will serve as a method reference for people interested in transcription errors. Based on their spectrum analysis, the authors speculate about different mechanisms by which these errors can be made. In order to be complete, the authors should analyze and report indels from their existing data.

We have now provided the analysis pipeline to identify indel errors in the transcriptome. The code can be found on Lynch-lab GitHub site. Estimates of indel error rates are 0.1-0.2 of those of base substitution rates. The results have been included as Supplementary file 1.

2) Concerning the model to explain the biased distribution of nonsense errors in mRNA, they proposed an NMD mechanism not yet characterized in bacteria, but this can easily be explained by transcription polarity described first by Morse and Yanofsky, (1969).

The polarity described by Morse and Yanofsky is defined as “the reduction in the relative rates of synthesis of those polypeptides specified by the genes of an operon on the operator-distal side of a mutated gene with a mutated gene with a nonsense codon”. Morse and Yanofsky explicitly excluded the arrested transcription mechanism for polarity and suggested that the polarity results from degradation of the untranslated region of RNAs that are immediately distal to a nonsense codon and are not covered by ribosomes. Therefore, the transcription polarity and the ribosomerelease model essentially refer to the same underlying biological process. The transcription polarity in Morse and Yanofsky, 1969 emphasizes the destabilizing effect of genetic polar mutants and it is not widely interpreted as a quality-control mechanism for mRNAs.

In the present study, we interpreted the biased distribution of nonsense errors as a result of a potential NMD-like mechanism not only because of the biased distribution of nonsense transcript errors in mRNAs, but also because of observations shown in Table 1, where the expected percentages of nonsense errors tend to be higher than the observed ones.

We also clarified in the manuscript that the ribosome-release model was firstly proposed by Brogna and Wen, 2009 as a speculative model for NMD. We believe both previous studies (Figure 3CD and Table 1. Entry 14,15 of Baker and Mackie, 2003; Figure 3 of Braun et al., 1998; Figure 2A of Nilsson et al., 1987) and our survey provide empirical evidence for this model and therefore highlight the existence of potential NMD-like mechanism in prokaryotes.

3) In order to make the sequencing method user-friendly, I would strongly suggest that the authors include a flowchart for the analysis of the sequencing data as they have done for the technical concept in Figure 1—figure supplement 1.

We agree. One of the goals of this study is to serve as the method reference for the transcript error community. We have now included a flowchart in Figure 1—figure supplement 2 to show how CirSeq reads are processed. Crucial steps such as how consensus sequences are constructed and reorganized are illustrated in detail.

Statistical analyses:4) While claims based on comparing the magnitudes of correlation coefficients of different binned data sets has now been removed, correlation coefficients for data that is not just binned (frequency data), but also non-independent (sliding windows) continue to be reported in Figure 4, Figure S2, and Figure S3. Corresponding regression models on these non-independent sliding window datapoints are inappropriately used to generate p-values. An alternative and somewhat less flawed approach is given only in the response to reviews document and not in the manuscript itself. This is presented as being the method of Meer et al.,( "with minor modifications", although they are not that minor, as they sacrifice the main advantage of that method. A good statistical model should have a good model of the error function. Count data has a binomial error function, well approximated as a Poisson. The advantage of the method of Meer et al., is that it exactly models this Poisson, but the method is dismissed in the response to reviews because it explicitly accounts for the fact that the raw count data depends on the number of reads, as though this were a bad thing. The alternative presented in the response to reviews does avoid the non-independence problem of the sliding window, but it still assumes homoscedasticity with respect to #errors/#reads, an assumption that is violated in practice given many more reads at some sites than others. If the authors want to model Equation (1) instead of Equation (2) (as numbered in the response to reviews), I would find this acceptable if and only if they (a) use a weighted regression to account for the known heteroscedasticity (ie at least model the magnitude of the error even if they don't model its shape), and (b) place this analysis in the manuscript itself as a replacement for doing a regression on the output of non-independent sliding-window-bins. The sliding window is perfectly appropriate and indeed very helpful as a visualization of the data, but inappropriate for the fitting and testing of statistical models.

Here the reviewer has two major comments. One is on the sliding-window analysis of binned data, and the other suggests accounting for sampling errors by using a weighted regression approach.

In response to the first major comment, we agree with the reviewer that the correlation coefficient from binned data should be avoided in evaluating the correlation between the frequency of nonsense errors and the distance between the nonsense error and the position of the original stop codon in a single base-resolution. In the current version of manuscript, we did the sliding-window (99% overlap of neighboring windows) analysis only for data visualization.

As the reviewer suggested, sampling errors should be considered in the regression analysis. We have updated the method by using a weighted regression method to evaluate whether there is a correlation between the frequency of a nonsense error and its distance from the original stop codon. The weight was calculated as the reciprocal of the variance of the nonsense error frequency, where the variance can be estimated as p(1−p)R. 𝑅 refers to the number and of ribonucleotides assayed (the read coverage), and 𝑝 is the estimated frequency of transcript errors.

The statistics from the weighted linear regression indicates that the negative correlation revealed by the sliding window analysis is significant in *E. coli*, *B. subtilis*, and *M. florum*, but not in *A. tumefaciens*. Compared to other three species, a smaller number of nonsense errors were detected in *A. tumefaciens* (59 vs. 109, 113, 105 in other three species), which may result in a low statistical power to reveal a potential pattern for the distribution of nonsense errors at a single base-resolution. This speculation is supported by our previous study (https://advances.sciencemag.org/content/3/10/e1701484, Figure 4F), where an increased number of nonsense errors detected in yeast transcription fidelity mutants vs. the wild type helps to reveal the negative correlation.

5) I take issue with the claim in subsection “A global view of the transcript error distribution” that transcript errors are randomly distributed across genes. After a Bonferroni correction for the number of genes, the manuscript has very little power with which to make such a claim of evidence of absence. What is more, Meer et al., showed (using a different *E. coli* dataset) that the error rate does depend on the strength of selection on the gene in question, as assessed by its protein abundance. This analysis had much higher power than the current study, firstly because there was only one degree of freedom and hence no need for a radical Bonferroni correction, and secondly because explicit modeling of the error function as described above will generally increase power. Given that (uncited) evidence that contradicts the manuscript's claim for randomness has already been published, and the lower power of the current manuscript's basis for making this claim, a claim of randomness should not be made unless the authors are able to apply the same or similarly high-powered method and directly show contradictory results. Should the previously published claim non-randomness be supported, this would have implications for the authors' model of randomly generated transcript error, eg subsection “Characterization of transcript errors”. In this passage too, the statistics are problematic. Observed and expected "data" are compared using a paired t-test, as though the expectations were themselves data with error magnitudes homoscedastic with those of the observed data.

We think this comment comes from the discrepancy between the argument of gene-specific transcript-error rates from Meer et al., 2019 and our observation of randomly distributed transcript errors. In response to this comment, we would like to first revisit observations and the corresponding conclusion from Meer et al., 2019. By analyzing a different *E. coli* dataset from Traverse and Ochman, 2016, Meer et al. claim an inverse correlation between protein abundance and non C-to-U transcript-error rates and argue for potential gene-specific transcript-error rates shaped by selection. Because the C-to-U error rate in the previous study is likely overestimated owing to technical errors, it is unclear whether the Meer et al., conclusion holds for the overall error rate. It is also unclear whether the gene-specific error rate argument applies for prokaryotes other than *E. coli*.

In the present study, we provided a per-gene approach to evaluate the distribution of transcript errors, where the sampling error related to the coverage of genes has been considered as Poisson distributed. Besides, we also ran a per-site analysis suggested by the reviewer. Results from both analyses indicate, in general, there is no sign for gene-specific transcript-error rates.

We have also updated the statistics and used chi-square tests to compare expected and observed percentages displayed in Table 1.

6) I am also confused by the bootstrapped standard deviation in Figure 4, Figure S2, and Figure S3. In Figure 4, why bootstrap rather than use the known formula for a binomial? In Figure S2 and Figure S3, why bootstrap only the numerator and not the denominator of the quantity whose error is being assessed?

We agree with the reviewer that the number of transcript errors observed at one locus is expected to follow a binomial distribution. Therefore, we have updated those figures and calculated the binomial standard deviation of error frequencies according to the read coverage and the frequency of transcript errors.

7) I still fear that the casual reader will take away from the Discussion section that differences between the two studies in error rates other than C->U are trivial in magnitude, which is not the case.

We agree with the reviewer that although the most striking differences comes from the C-to-U error rate, other types of substitution rates are also different between the present study and Traverse and Ochman’s study. We have now clearly pointed out this difference in the main text. It is possible that metal ion-based fragmentation may also induce non C-to-U error rates.

We also would like to point out that we provided details on our data and analysis methods, such as the strain number, growth phase, and criteria to filter false positives, which we think may be helpful for future studies to compare with our work.